# Investigation on Chalcogenide Glass Additive Manufacturing for Shaping Mid-infrared Optical Components and Microstructured Optical Fibers

**Julie Carcreff [1], François Cheviré [1], Ronan Lebullenger [1], Antoine Gautier [1], Radwan Chahal [2], Jean Luc Adam [1], Laurent Calvez [1], Laurent Brilland [2], Elodie Galdo [1], David Le Coq [1], Gilles Renversez [3] and Johann Troles [1,*]**

[1] CNRS, University of Rennes, ISCR-UMR 6226, 35000 Rennes, France; julie.carcreff@univ-rennes1.fr (J.C.); francois.chevire@univ-rennes1.fr (F.C.); ronan.lebullenger@univ-rennes1.fr (R.L.); antoine.gautier@univ-rennes1.fr (A.G.); jean-luc.adam@univ-rennes1.fr (J.L.A.); laurent.calvez@univ-rennes1.fr (L.C.); elodie.galdo@univ-rennes1.fr (E.G.); david.lecoq@univ-rennes1.fr (D.L.C.)

[2] Selenoptics, 263 Avenue Gal Leclerc, 35042 Rennes, France; radwan.chahal@selenoptics.com (R.C.); laurent.brilland@selenoptics.com (L.B.)

[3] CNRS, Centrale Marseille, Institut Fresnel, Aix-Marseille University, UMR 7249, 13013 Marseille, France; gilles.renversez@fresnel.fr

\* Correspondence: johann.troles@univ-rennes1.fr; Tel.: +33-(0)2-23-23-67

**Abstract:** In this work, an original way of shaping chalcogenide optical components has been investigated. Thorough evaluation of the properties of chalcogenide glasses before and after 3D printing has been carried out in order to determine the impact of the 3D additive manufacturing process on the material. In order to evaluate the potential of such additive glass manufacturing, several preliminary results obtained with various chalcogenide objects and components, such as cylinders, beads, drawing preforms and sensors, are described and discussed. This innovative 3D printing method opens the way for many applications involving chalcogenide fiber elaboration, but also many other chalcogenide glass optical devices.

**Keywords:** chalcogenide glasses; 3D printing; mid-infrared fibers; photonic crystal fibers

## 1. Introduction

In recent years, a growing interest has been developed for optical materials and fibers for the mid-infrared (mid-IR) region, especially for chalcogenide glasses [1,2]. Such interest originates from societal needs for health and environment for instance, and also from the demand for military applications [3,4]. Indeed, the mid-IR spectral region contains the 3–5 µm and 8–12 µm atmospheric transparent windows, where both military and civilian thermal imaging can take place [5,6]. Compared to oxide-based glasses, vitreous materials containing chalcogen elements, i.e., S, Se and Te, show large transparency windows in the infrared. Depending on their chemical composition, chalcogenide glasses can be transparent from the visible up to 12–18 µm [7–9]. In this context, we have investigated an alternative approach based on a 3D printing process for fabricating mid-IR optical components such as preforms, optical fibers, sensors and beads.

3D printing, more formally known as additive manufacturing, was firstly a shaping technique widely used for polymer for rapid prototyping processes but it is now more and more commonly employed for mass production and mass customization, as well [10–13]. Very quickly, this technology was extended to metals [14], ceramics [15] and even glasses [16–18]. The most common method for polymers is the fused filamentation fabrication (FFF) method, also known as fused deposition modeling (FDM) [19,20]. It involves the extrusion of a thermoplastic material through a heated die and accurate deposition, layer by layer, until the desired three-dimensional object is obtained. It has been

demonstrated that such filamentation techniques can be modified and applied to the 3D printing of glasses with low glass transition temperature ($T_g$), also known as "soft glasses", such as chalcogenide glasses [21,22] and phosphate glasses as presented in reference [23]. In previous works, in reference [22], it has been shown that a microstructured optical fiber can be obtained from a 3D printed chalcogenide glass preform. The present study is a more complete investigation of the thermal and optical properties of the chalcogenide glasses after the 3D printing process.

For this work, the 3D-printing set-up is based on a customized desktop RepRap-style 3D printer running Marlin firmware [24]. This printer, typically used with polymer filaments to produce plastic objects, was upgraded to reach a deposition temperature of up to 400 °C required for soft glasses and the feeding mechanism was customized to suitably handle brittle materials such as glasses.

Different physical and optical properties of the printed glasses such as density, thermal expansion coefficient, refractive indices, and transmission have been investigated and compared to the properties of classical melt-quenched glasses. By using this additive manufacturing method, chalcogenide cylinders, pellets, beads, as well as microstructured performs with complex designs can be fabricated in a single step with a high degree of repeatability and an accuracy of the geometry. This original 3D printing method opens the way for numerous applications, involving chalcogenide fiber manufacturing, but also many other chalcogenide glasses optical devices.

## 2. Materials and Methods

### 2.1. Thermal and Physical Optical Properties of the Chosen Glass: $Te_{20}As_{30}Se_{50}$

The selected glass for printing shall present thermal properties compatible with manufacture by fused filamentation fabrication. To begin, the use of a commercial extrusion head usually utilized for conventional thermoplastics such as polylactic acid (PLA) requires that the selected glass shows a low glass transition temperature ($T_g$). In addition, it would be preferable that the glass is stable with respect to crystallization. In this context, the $Te_{20}As_{30}Se_{50}$ (TAS) chalcogenide glass that exhibits a $T_g$ slightly below 140 °C appears to be a good candidate [25]. Figure 1a shows the differential scanning calorimetry (DSC) curve of the TAS glass measured with a DSC Q20 from TA instrument with a heating rate of 10 °C/min. The glass exhibits a $T_g$ close to 137 °C and no sign of crystallization are observed up to 300 °C, which confirms that the glass could be compatible with the FFF 3D printing process.

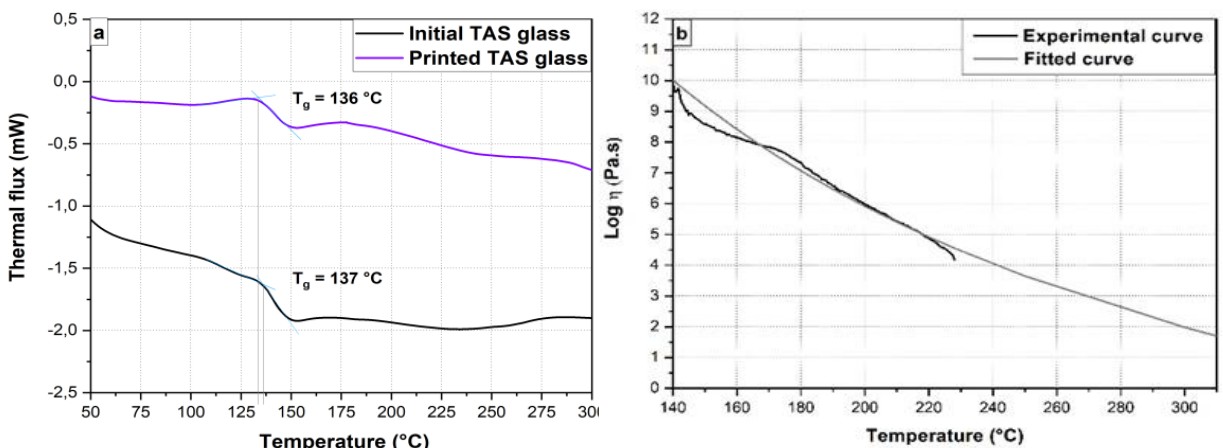

**Figure 1.** Thermal properties of the $Te_{20}As_{30}Se_{50}$ (TAS) chalcogenide glasses: (**a**) differential scanning calorimetry measurement, (**b**) viscosity versus temperature curve.

The implementation of a filamentation process with this glass also requires a better knowledge of its viscosity as a function of temperature. For this purpose, a TAS pellet was

prepared in order to measure the viscosity. Both sides of the printed disk (15 mm diameter, 4 mm height) were polished to ensure that they were parallel. The viscosity of the glass was measured above the glass transition temperature between 140 and 230 °C by using a Rheotronic®parallel plate viscometer (Theta industries). This experimental set-up did not permit one to measure viscosities lower than $10^4$ Pa.s. This value is reached when the temperature of the glass is equal to 230 °C. However, considering that experimental data follow the VFT (Vogel–Fulcher–Tammann) law, the viscosity value can be estimated up to 300 °C using a fitting procedure from Equations (1) and (2) [26], as shown in Figure 1b. This enables one to estimate the flow rate and, therefore to optimize the process.

$$\log_{10} \eta(T) = \log_{10}\eta_\infty + \left( \frac{A}{T - T_0} \right) \tag{1}$$

This equation can also be written as follows:

$$\log_{10} \eta(T) = \log_{10}\eta_\infty + \frac{\left(\log_{10} \eta(T_g) - \log_{10} \eta_\infty\right)^2}{m\left(\frac{T}{T_g} - 1\right) + \left(\log_{10} \eta(T_g) - \log_{10} \eta_\infty\right)} \tag{2}$$

where, $\log_{10}\eta_\infty$, m and $T_g$ are constants determined by the fitting procedure. They are equal to $-9$, 35.9 and 413.15 K, respectively.

Commonly, the required viscosities for polymers in FFF processes are around $10^2$–$10^3$ Pa.s [27,28]. According to Figure 1b, such viscosities are obtained when the temperature gets to the 270–300 °C range, which confirms that the TAS glass is a suitable candidate for FFF printing at reasonable temperatures.

The optical properties of the glass also have to be studied to ensure its suitability for the targeted infrared optical applications, particularly for the realization of infrared components and fibers.

The transmission of a polished 1-mm thick TAS glass pellet was measured with a Fourier Transform Spectrometer (FTIR) (Bruker Tensor 37) from of 1 μm to 22 μm. The obtained transmission spectrum is shown in Figure 2. The transmission of the glass ranges from 1.2 to 20 μm, with a maximum of transmission of around 65%. Such moderate transmittance is explained by strong Fresnel reflections due to the high refractive index of TAS glass. Indeed, as presented in Figure 2, the refractive index of the TAS glass is close to 3.

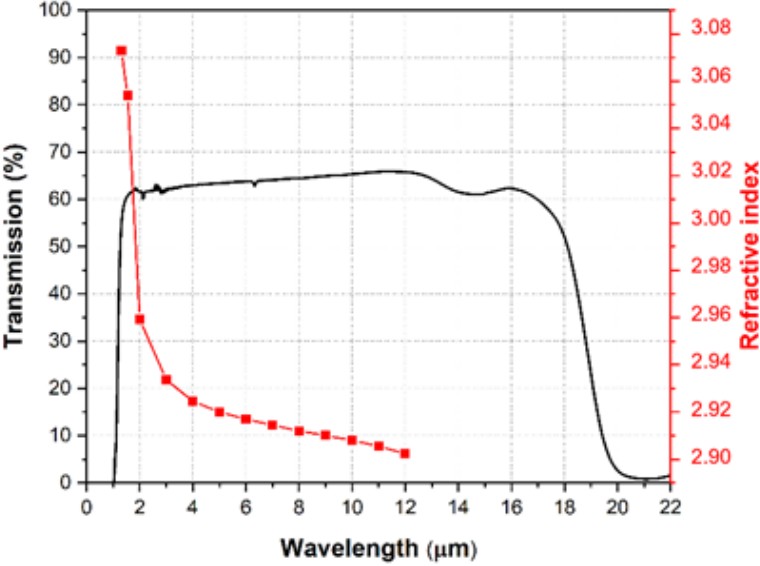

**Figure 2.** Transmission and refractive indices of $Te_{20}As_{30}Se_{50}$ chalcogenide glass in the mid-infrared spectral region

Refractive indices were measured for different wavelengths between 1.3 and 12 μm. From 2 to 12 μm, a homemade optical bench based on the minimum deviation method or Littrow method [29] was utilized by using a TAS prism [30,31]. Due to the experimental limits of our prism method (2–12 μm), two other measurements have been achieved at 1.31 μm and 1.55 μm with a commercial Metricon device (Model 2010/M). The refractive index of the TAS glass is equal to 3.073 at 1.31 μm and decreases to 2.902 at 12 μm.

## 2.2. Additive Manufacturing Process

In this study, the 3D-printing set-up is based on a customized commercial RepRap-style 3D printer (Anet A8) upgraded for soft glasses. The feeding mechanism is especially customized for brittle materials. The extrusion head, made of copper, can reach temperatures up to 400 °C. As shown in Figure 2, TAS glass reaches the appropriate viscosity when the temperature is above 270 °C. As a result, the temperature of the printer head was set to around 300 °C. The width of the printed lines resulted from the size of the nozzle diameter, which was mainly chosen at 400 μm. However, we have shown that the printing process can also use a 250 μm diameter nozzle. The chalcogenide glass was deposited on a sodalime silicate glass bed heated to 140 °C, which ensured the good adhesion of the printed sample. It can be noted that all the components printed in this study were extruded using a layer thickness of 100 μm, which corresponds to a z-pitch of 100 μm for each layer.

The 3D printer was fed by 3 mm diameter TAS rods which were prepared from TAS bars synthesized by the melt–quenching method. To save time and to ensure that the printing process was possible with such a glass, the preliminary tests were performed with unpurified TAS glasses. The different chemical elements, i.e., Te (5N), As (5N) and Se (5N), were placed in a silica ampoule under a vacuum. The ampoule was then heated at 850 °C in a rocking furnace for a few hours and finally the molten glass was quenched at room temperature and placed in an annealing furnace at $T_g$ +5 °C for 3 h and then slowly cooled to ambient temperature.

After the validation of the additive manufacturing method for TAS, the raw material rods were made from purified glass using a similar process to the one described in reference [32]. In this case, $TeCl_4$ and Al (1000 ppm and 100 ppm, respectively) were added during the glass synthesis. After the first melt–quench, the glass was distilled for the first time under a dynamic vacuum to remove impurities such as $CCl_4$ and HCl. The second stage of static vacuum distillation made it possible to remove carbon, silica, alumina and other refractory oxides, for example. The glass was then homogenized at 850 °C for a few hours, then cooled to 550 °C and quenched in water before it was annealed at $T_g$ +5 °C.

Once removed from the synthesis silica tube, the TAS glass bars (also named TAS preforms) were placed in an annular furnace within a homemade fibering tower to be drawn into 3 mm diameter rods. Whether it was an unpurified or purified glass preform, meters of fibers were also drawn in order to measure the optical transmission of the initial glasses that would be used to feed the 3D printer (the details of these characterizations will be presented in Section 3.2.). As an example, a 12 cm long glass bar gives about 2 meters of TAS rod.

The rods were then fed into the printer's head to be extruded. The printer movements were controlled by using G-code programs especially computed for driving displacements compatible with TAS glass. Firstly, as presented in Figure 3, simple objects such as cylinders, disks, and beads were made to study the impact of the 3D printing process on the properties of TAS glass such as density, thermal expansion coefficient, refractive indices and optical attenuation. In a second step, more complex shapes were realized such as microstructured preforms (studied in [22] and Section 3.3) and tapers.

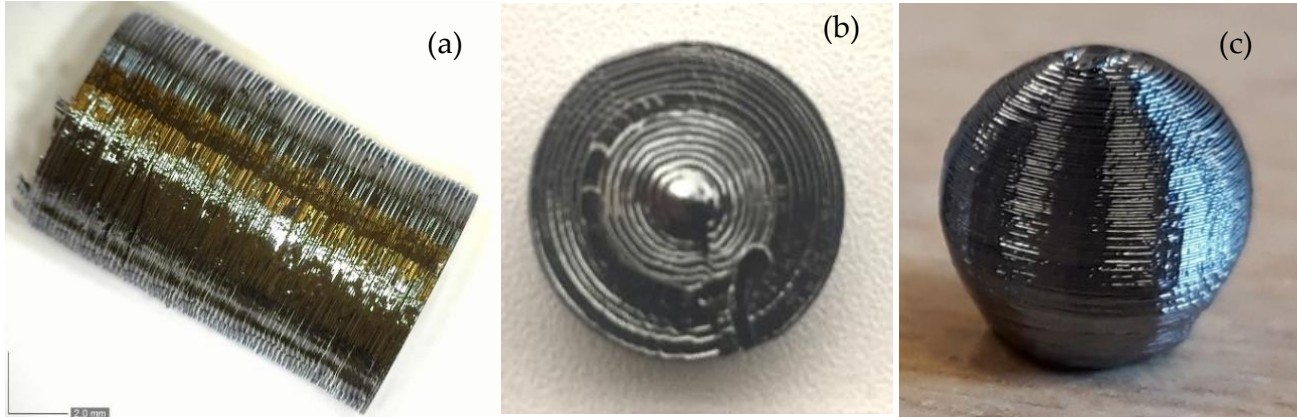

**Figure 3.** Chalcogenide 3D printed samples: (**a**) chalcogenide cylinder, (**b**) chalcogenide disk (**c**) chalcogenide bead.

## 3. Results

### 3.1. Physical Properties of the Printed Glasses

A 5 mm high TAS pellet with a 15 mm diameter was printed as shown in Figure 3b, and polished before being analyzed to compare the physical properties of the initial glass to those of the printed glass. The composition of the pellet was determined by energy dispersive spectroscopy (EDS). The results are shown in Table 1. The comparison with the initial composition of the glass indicates that there is no change in the chemical composition during the printing process. The glass transition temperature ($T_g$), thermal expansion coefficient ($\alpha$), density and X-ray diffraction pattern of the printed TAS glass were also compared to a standard TAS glass (see Table 1 and Figure 4). Considering the experimental error inherent to DSC measurements (DSC Q20 Thermal Analysis), the $T_g$ of the printed glass and the initial glass are similar. In addition, no crystallization peak was observed in the DSC measurement of the printed glass (not shown). The thermal coefficient measurements were carried out on a thermomechanical analyzer (TA Instrument, TMA 2940) with a heating ramp of 2 $^\circ$C·min$^{-1}$. The coefficient is 22.82·10$^{-6}$ K$^{-1}$ for a classic glass and 22.62·10$^{-6}$ K$^{-1}$ for a printed glass, respectively. Considering the measurement error, both glasses have the same coefficient of expansion. The densities were measured by the Archimedes method on a Mettler Toledo XS64 balance. It is noticed that the density significantly decreases from 4.86 g·cm$^{-3}$ for a quenched glass to 4.81 g·cm$^{-3}$ for a printed glass, meaning that the printed objects do not present the expected density of a bulk glass.

**Table 1.** Physical property comparison of a melt/quenched glass used as the initial glass and a printed glass.

|  | Initial Glass | Printed Glass |
| --- | --- | --- |
| Composition from EDS analysis (%) ($\pm$1) | $Te_{20}As_{30}Se_{50}$ | $Te_{21}Se_{29}Te_{50}$ |
| Bulk transmission range (µm) | 1–18 | 1–18 |
| Fiber transmission range (µm) | 2–12 | 2–12 |
| $T_g$ ($^\circ$C) ($\pm$2) | 137 | 136 |
| Coefficient of thermic expansion (10$^{-6}$ K$^{-1}$) ($\pm$ 1%) | 22.82 | 22.62 |
| Density (g·cm$^{-3}$) ($\pm$ 0.02) | 4.86 | 4.81 |

Finally, to ensure that the glass did not crystallize during the additive manufacturing process, the initial and the printed TAS pellets were analyzed by X-ray diffraction (XRD). XRD diagrams were recorded at room temperature in the 10–80° 2θ range using a PANalytical X'Pert Pro (Cu $K_{\alpha1}$, $K_{\alpha2}$ radiations, $\lambda_{K\alpha1}$ = 1.54056 Å, $\lambda_{K\alpha2}$ = 1.54439 Å, 40 kV, 40 mA, PIXcel detector 1D). Figure 4 shows that the two diagrams perfectly overlap and no additional diffraction peak is detected for the printed glass.

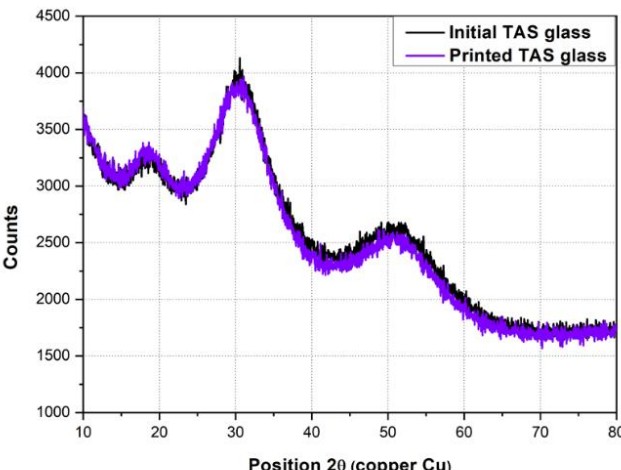

**Figure 4.** XRD diagrams of initial TAS glass (black curve) and printed TAS glass (purple curve).

### 3.2. Optical Properties of the Printed Glasses

In order to study the consequences of 3D printing on optical properties, 10 mm diameter chalcogenide beads were 3D printed using concentric extruded layers as illustrated in Figure 5a. Two beads were polished to obtain 1 mm-thick pellets, one along the XY plane and the other one along the XZ plane. The latter, from the process aspect, is equivalent to the plan YZ. It is important to remember that the Z axis is the axis of the layer stacking. The transmission windows of the pellets recorded on a Fourier Transform InfraRed Spectrometer (FTIR, Bruker Vector 22), in the 1.25–22 μm range are shown in Figure 5b. The transmission spectra of the printed and initial TAS glasses are similar in terms of spectral range. However, the maximum transmission of the extruded TAS is limited to 48% for the XY plane and 39% for the XZ plane, compared to 65% for the initial glass.

To observe the effect of the successive addition of the different layers and lines, images were recorded with an infrared camera working in the 7–13 μm wavelength windows (FLIR A655sc). The black frame photograph in Figure 5c corresponds to a TAS pellet made from a melt–quenched glass (used as the initial glass for printing), which seems to be perfectly homogeneous in the infrared range from 7 to 13 μm. The other two photographs are those of the glass pellets printed either in the XZ plane for the blue frame or in the XY plane for the red frame. Imperfections are present in both pads. For the XZ plane, it looks like a periodic accumulation of points attributed to the cross-section of the printed lines while for the XY plane the defects are placed in a circular way corresponding to the in-plane printed lines.

After pellets and beads, two cylinders (or preforms) with an outer diameter of 8 mm were printed. One of the cylinders was obtained from an unpurified TAS glass rod while the second one was obtained from a purified glass rod. These preforms were prepared in order to be drawn and optically characterized with the aim to compare the optical losses between initial glass fibers and printed glass fibers depending on the quality of the initial glass.

Each preform was stretched on a drawing tower specifically designed for low $T_g$ glasses. Several meters of fibers with approximately a 400 μm diameter were obtained and optically characterized using the FTIR and a MCT (alloy of mercury, cadmium and tellurium) detector. Attenuation measurements were performed in order to compare the optical transmission of the initial glass fibers and the "printed" fibers. The results are shown in Figure 6. A significant increase in optical losses is observed for the "printed" fibers. Thus, when the minimum attenuation of the initial unpurified glass is less than 10 dB/m, and less than 1 dB/m for the purified glass, the minimum attenuation of the unpurified "printed" fiber is around 28 dB/m and around 18 dB/m for the purified "printed" fiber.

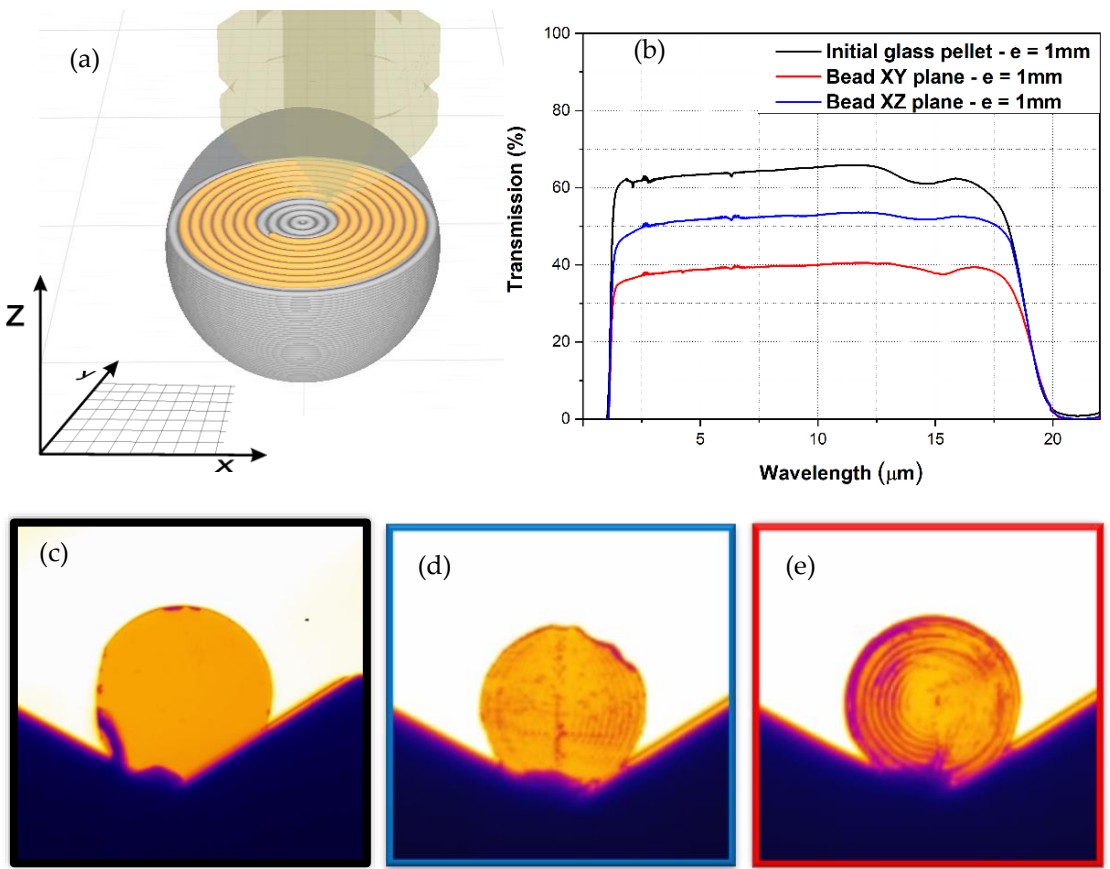

**Figure 5.** Optical transmission of printed beads: (**a**) numerical view of the 3D printing process, (**b**) initial glass transmission compared to the transmission of 2 disks cut in the XY plan (red curve) and XZ plan (blue curve), respectively. IR camera views in the 7–13 μm window (**c**) initial TAS glass, (**d**) XZ plan, (**e**) XY plan.

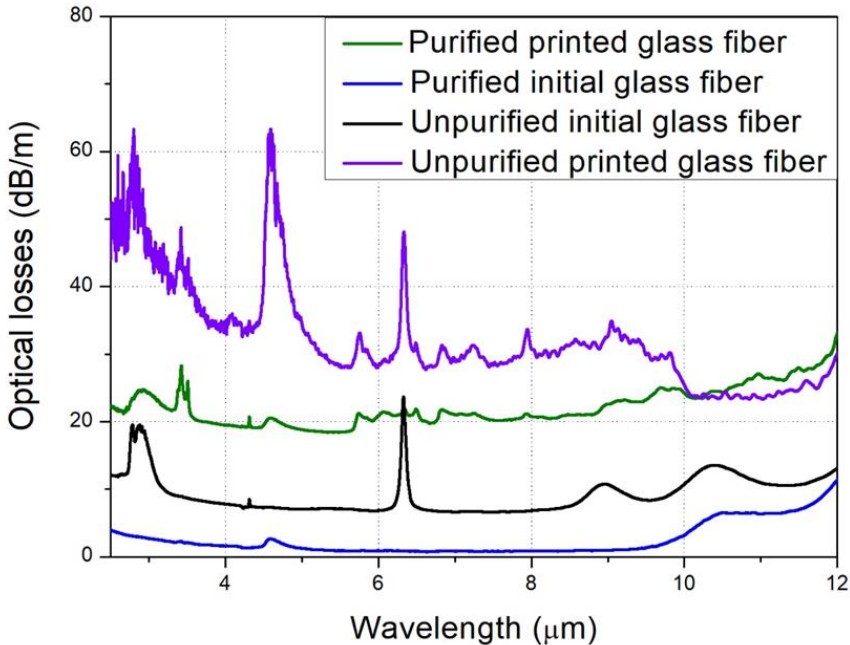

**Figure 6.** Attenuation spectra of the initial glasses (unpurified and purified glasses, red curve and blue curve respectively) and the "printed" glass fibers (from an unpurified glass and from a purified glass, purple and green respectively)

### 3.3. Optical Components and Fibers Printing

The printed objects used for the previous characterizations were bulk in nature with a concentric shape. It is therefore interesting to evaluate how the chalcogenide glass behaves when the printed objects are thinner and mainly made up of a straight line, for potentially making smaller components, such as sensors [33,34] by using this innovative 3D printing process.

First, a slot shape object has been printed, as shown in Figure 7a. The lines are around 400 μm wide and 700 μm high. The width corresponds to the diameter of the nozzle (400 μm) and the height is the sum of seven printed layers (7 × 100 μm). The longest lines are equal to 3 cm and the shortest to 1 cm. The second object, as shown in Figure 7b, is a glass comb whose height is around 1 mm, the width of the teeth being close to 800 μm, which corresponds to two printed lines. The length of the teeth is 3 cm.

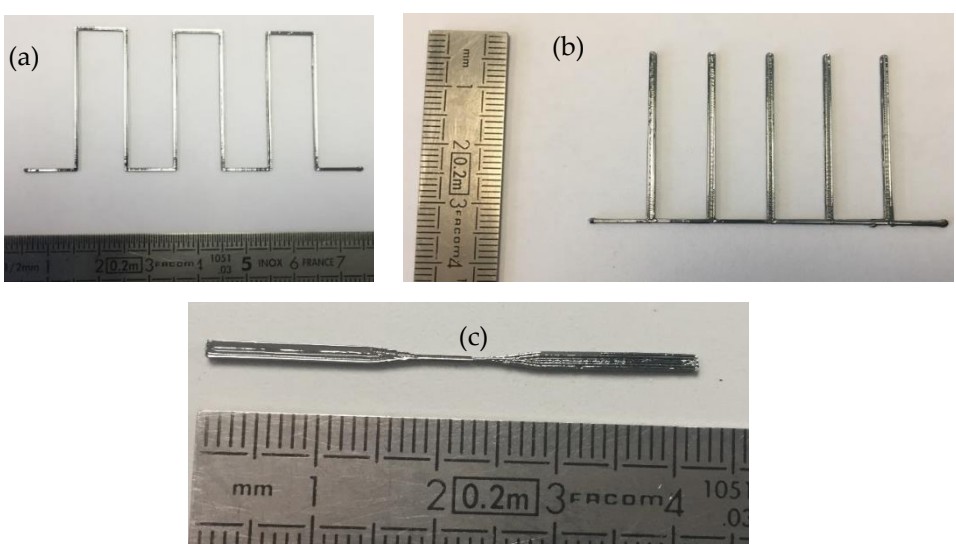

**Figure 7.** Linear 3D printed chalcogenide glass: (**a**) 700 × 500 μm slots shapes, (**b**) 1000 × 800 μm comb and (**c**) 1000 × 250 μm rectangular taper.

A third object was printed with a 0.25 mm diameter nozzle using the same thermal conditions and feeding rods. It is a straight object with two wide edges of 1000 × 1000 μm² over a length of 15 mm connected by a taper of 1000 × 250 μm² over a length of 10 mm, as shown in Figure 7c. For these small printed components, variations of about 10% in height and width were measured. These results show that various shapes of chalcogenide glass objects can be obtained by additive manufacturing and demonstrate the strong potential of such a fabrication approach.

Finally, additive manufacturing was applied to the printing of fiber preforms as previously shown in reference [22], with the printing of a hollow-core preform with a clad and six capillaries. After this first proof of concept, more complex designs were printed as shown in Figure 8a. The first preform is a hollow-core preform constituted by 8 capillaries of 1.2-mm diameter. The preform had an outer diameter of 18 mm and a height of 15 mm. This preform was too short to be drawn. A second preform, illustrated in Figure 8b, has been drawn into fiber (Figure 8c). This printed preform was designed with eight half-capillaries as illustrated in Figure 8b. The printing of a such geometry permits one to reduce the diameter of the preform and consequently permits one to increase the printed preform length by using the same volume of glass. The diameter of the semi-circles is 1.8 mm. The overall diameter and height of the preform are 15.2 mm and 30 mm, respectively. This design was realized also for showing the interest of 3D printing that can permit one to obtain geometries that cannot be obtained by the stack and draw technique. However, no light propagation in mid-IR was observed in the hollow core of the fiber drawn from this preform. This is attributed to the final geometry of the as-prepared fiber (Figure 8c)

that needs to be optimized to allow mid-IR light transmission (core size, half-capillaries thickness . . . ).

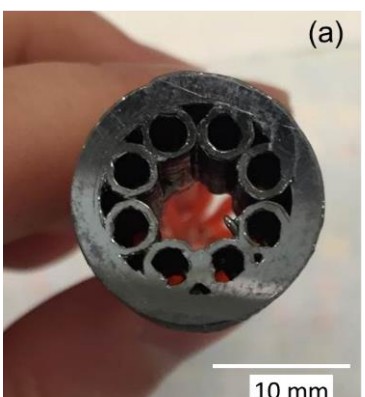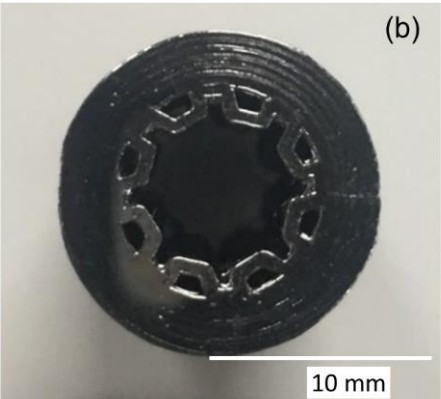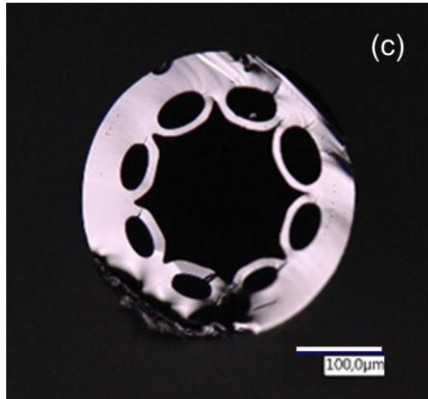

**Figure 8.** Printed preforms: (**a**) cross-section view of the chalcogenide printed 8 capillaries preform and (**b**) cross-section view of the chalcogenide printed 8 half-capillaries preform (**c**) cross section of a hollow core fiber drawn from the preform shown in (**b**).

## 4. Discussion

The aim of this work was, firstly, to study the impact of 3D printing by filamentation on the properties of chalcogenide glasses, and secondly, to apply additive manufacturing to produce optical objects and components for mid-infrared applications.

The comparison of the physical and thermal properties between the initial glass and the printed glass, as presented in Table 1, shows no major differences. However, the investigation of the optical properties showed a significant difference between the two processes. Indeed, a significant decrease in transmission was observed both on bulk printed glasses and printed fibers. The transmission of TAS glass obtained by the usual melt–quench process exceeds 60% in the 2–18 μm window, whereas the transmission along the Z axis, which corresponds to the axis of stacking of the layers, is below 40%, and the transmission in a direction perpendicular to the Z axis is 50%. As illustrated in Figure 5d,e, the images recorded with a thermal camera in both directions show the presence of numerous inhomogeneities and scattering defects which are responsible of the decrease in transmission. These inhomogeneities could be refractive index variations or additional absorptions at the interfaces between the printed lines and the printed layers, and could also be the results of small bubbles trapped within the glass, as already observed in previous work [22]. The presence of small crystals in the printed pieces cannot be totally excluded, even though XRD investigations have shown no sign of crystallization due to the process (Figure 4). The infrared images also show that the interfaces between the lines (along the Z axis, Figure 5e) present more defects than the interfaces between the layers (Figure 5d). Indeed, the IR image, presented in Figure 5e, clearly shows concentric lines that are related to the movement of the extrusion head for one printing layer (Figure 5a). In order to improve the transmission of the printed chalcogenide glasses, this effect should be considered in further studies, by changing the 3D printing parameters such as the nozzle size, the layer thickness, the extrusion temperature, the extrusion, and printing rates, which have not yet been investigated.

The study of the optical properties of fibers has also revealed significant differences between the initial glass and the printed glass and in particular shows the impact of the quality of the starting glass. The minimum of attenuation of the initial glass were measured at around 8 dB/m and 1 dB/m for the unpurified glass fiber and purified glass fiber, respectively. The minimum of attenuation of the "printed" fiber made from the unpurified glass is close to 28 dB/m, whereas the preform realized with a purified glass permit one to obtain propagation losses in the fiber lower than 18 dB/m. So, the quality of the printed glass is clearly better when a purified glass is used for printing. However, it can be noticed

that the additional losses induced by the printing process are similar. Indeed, the additional losses induced by the 3D printing is 20 dB/m for the unpurified glass and 17 dB/m for the purified glass, which indicates that the majority of the losses are induced by the 3D printing process itself. In addition to the background losses, numerous other additional absorption peaks are observed in the attenuation spectra of both printed glasses drawn into fibers: OH groups and molecular $H_2O$ signatures at 2.9 μm and 6.3 μm respectively, C–H chemical IR signatures at 3.3 μm, other undefined organic compounds signatures at 5.8 μm, 6.8 μm and 7.3 μm, and finally oxidation signatures at 7.9 at 9.1 μm. It is important to note here that the 3D printing process is carried out in air. In these experimental conditions, the origin of the additional absorption bands is probably due to the presence of moisture, oxygen and possible traces of pollution by organic compounds in the printer's enclosure. For the fiber drawn with an unpurified printed preform, the strong increase in the Se–H bond IR signature at 4.55 μm is surprising in comparison to the purified glass fiber spectrum for which the characteristic band of Se–H does not evolve. However, Se–H bonds are the result of a chemical reaction between chalcogenide glass and hydrogen coming mainly from O–H bonds and/or molecular water. In our case, before printing, the unpurified initial glass shows high O–H and molecular water contents, at 2.9 μm and 6.3 μm respectively, whereas the purified glass does not contain any water pollution. The most probable hypothesis is that the water contained in the initial unpurified glass has reacted with the selenium of the glass to form Se–H bonds during printing.

The main interests of additive manufacturing processes are the realization of complex objects that cannot be obtained by other methods but also for mass production and for mass customization with a significant cost reduction. In order to evaluate the potential of our 3D printed method, various shapes and small objects were achieved.

Concerning the objects showing centimetric sizes, it has been shown that several shapes such as disks, cylinders, and beads can be 3D printed with the selected chalcogenide glass. However, the investigations have shown that improvements must be implemented in order to reach the same optical transmission in a printed glass than in a glass synthesized by the usual melt–quenching process. For future studies, the priority improvement should be the use of an inert and dry atmosphere in the printed enclosure to prevent any defects resulting from chemical pollution.

In the present experimental set-up, the resolution of one line (approximately equal to the size of nozzle) is 250 μm or 400 μm and the resolution along the Z axis is 100 μm, which corresponds to the thickness of one layer. For preliminary results, these resolutions seem to be sufficient for printing components with size smaller than 1 mm. These preliminary results also show that it will be possible to exploit the properties of chalcogenide glasses to make infrared sensors. In fact, the printed taper in Figure 7c exhibits a design very close to that of infrared fiber sensors used commercially by DIAFIR company to carry out infrared spectroscopy and medical diagnosis [33,34].

## 5. Conclusions

The investigations on the thermal properties between the initial glass and the printed glass have shown no major difference. Indeed, the composition of the glass and the glass transition temperature are the same before and after the printing operation. In addition, no sign of a crystallization is reported in the printed glasses. The study has also shown that improvements have to be achieved in order to increase the optical quality of the printed glasses. For example, the use of an inert and dry atmosphere during the printing has been pointed out and this should constitute the main future development axis.

Finally, this work clearly demonstrates that 3D printing could be an innovative approach for elaborating microstructured fibers. Especially, it allows one to obtain new geometries that cannot be realized by any other methods such as stack and draw [35,36], extrusion [37] or molding [38].

This innovative 3D printing method opens the way to many applications involving chalcogenide fiber manufacturing, but also many other optical devices based on chalcogenide glasses, such as chalcogenide sensors for spectroscopy and medical diagnosis.

**Author Contributions:** Conceptualization, J.T., L.C., R.L., F.C. and G.R.; methodology, A.G., F.C., L.B. and J.C.; formal analysis, E.G. and J.C.; investigation, J.C., R.C. and L.B.; writing—original draft preparation, J.C.; writing—review and editing, J.T., F.C., D.L.C., G.R. and J.L.A.; supervision and project administration, J.T.; funding acquisition, J.T., G.R. and L.B. All authors have read and agreed to the published version of the manuscript.

**Funding:** This work was funded in part by the European Union through the European Regional Development Fund (ERDF), the Ministry of Higher Education and Research, the French region of Brittany, Rennes Métropole, and the French Délégation Générale pour l'armement (DGA) (grant ANR ASTRID DGA FOM-IR-2-20).

**Institutional Review Board Statement:** Not applicable.

**Informed Consent Statement:** Not applicable.

**Data Availability Statement:** The datasets generated during and/or analysed during the current study are available from the corresponding author on reasonable request.

**Acknowledgments:** See funding section.

**Conflicts of Interest:** The authors declare no conflict of interest.

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
