# Peer review of "Investigation on Chalcogenide Glass Additive Manufacturing for Shaping Mid-infrared Optical Components and Microstructured Optical Fibers"

_crystals, doi:10.3390/cryst11030228_

Round 1

Reviewer 1 Report

The paper report on innovative fabrication protocol of chalcogenide glass based microstructured optical fiber and in general optical components. The paper is well written and the topics realy innovative. I suggest to publish the paper

Author Response

Thank you for your review and your comments

Reviewer 2 Report

This is a review of “Investigation on chalcogenide glass additive manufacturing for shaping mid-infrared optical components and microstructured optical fibers,” by Carcreff et al. The authors describe a process for fabricating objects by 3D printing of Te20As30Se50 glass and provide measurement results for the glasses and printed objects. This is interesting work that is well-presented. I recommend publication, but there are a few points I think the authors should address.

  1. Lines 23-24: “Such interest originates from societal needs for health and environment for instance…” I don’t understand this statement. One criticism of chalcogenide glasses is that they are not environmentally friendly since they often contain arsenic.
  2. 1: The figure appears to be blurry, and the right side is cut off in the pdf I am looking at.
  3. Section 3.2: When measuring transmittance through the beads, doesn’t lensing take place? How does this affect the measurement?
  4. Section 3.3, printing fibers: The authors note that they some previously printed work on microstructured fibers in Ref. [22]. It would be helpful to include a more detailed explanation of how the current work is different from the previously published work.
  5. Lines 281-283: The authors note that the presence of inhomogeneities leads to scatter that explains the increased loss. I think this explanation is correct, but couldn’t loss also be due to increased absorption? Can the authors separate these effects? A scatter measurement could help rule out absorption.
  6. General: Have the authors explored annealing samples to reduce inhomogeneities and improve transmittance?

Minor points:

  • Line 14: Should be “means”, not “mean”.
  • Lines 14-16: Sentence starting “Moreover…” Check grammar.
  • Line 52-53: Should be “preforms” not “performs”.
  • Line 73: I think it should just be “viscosity as a function of temperature”, not “viscosity behavior”.
  • Line 142: Should be “meters of fiber”.

Author Response

This is a review of “Investigation on chalcogenide glass additive manufacturing for shaping mid-infrared optical components and microstructured optical fibers,” by Carcreff et al. The authors describe a process for fabricating objects by 3D printing of Te20As30Se50 glass and provide measurement results for the glasses and printed objects. This is interesting work that is well-presented. I recommend publication, but there are a few points I think the authors should address.

  1. Lines 23-24: “Such interest originates from societal needs for health and environment for instance…” I don’t understand this statement. One criticism of chalcogenide glasses is that they are not environmentally friendly since they often contain arsenic.

Applications in mid-IR spectroscopy of chalcogenide glass include detection of pollutants or gas like CO2, and for example developing medical diagnosis (see www.diafir.com). We agree that arsenic and arsenic oxides are toxics, however, As-containing chalcogenide glasses are less toxic than pure As or As2O3 (exactly like gallium arsenide in marketed smart phone). In, addition, a study done by an independent company, Biomatech, for the University of Rennes, has shown (on rats) that the TAS glass is biocompatible (ISO standard 10993-1). Of course, we agree, precautions must be taken during synthesis and disposal of the samples, like any other usual chemical product.

  1. 1: The figure appears to be blurry, and the right side is cut off in the pdf I am looking at.

The quality of the figure 1 has been improved

  1. Section 3.2: When measuring transmittance through the beads, doesn’t lensing take place? How does this affect the measurement?

The beads have been cut and polished in order to obtain 2 different disks (two beads are needed to characterized the two perpendicular planes). So, we don’t observe any lensing effects.

  1. Section 3.3, printing fibers: The authors note that they some previously printed work on microstructured fibers in Ref. [22]. It would be helpful to include a more detailed explanation of how the current work is different from the previously published work.

The results of the ref 22 is the proof of concept of using 3D printing for obtaining a chalcogenide PCF. In the present paper a more complete investigation of the printing process is presented.

This sentence has been added in the introduction of the article :

In previous works, in reference [22], It has been shown that a microstructured optical fiber can be obtained from a 3D printed chalcogenide glass preform. The present study is a more complete investigation of the thermal and optical properties of the chalcogenide glasses after the 3D printing process.

  1. Lines 281-283: The authors note that the presence of inhomogeneities leads to scatter that explains the increased loss. I think this explanation is correct, but couldn’t loss also be due to increased absorption? Can the authors separate these effects? A scatter measurement could help rule out absorption.

Right, If we take into account the presence of additional absorption bands observed on fibers at 3.3 µm, 5.8 µm, 6.8 µm, 7.3, 7.9 and 9.1 µm, additional absorption of the defects can not be excluded. However, our Mid-IR imaging set-up cannot discriminate the two contributions.

So, this sentence has been modified in the text  :

These inhomogeneities could be refractive index variations or additional absorptions at the interfaces between the printed lines and the printed layers, and could be also the results of small bubbles trapped within the glass, as already observed in previous work [22]

  1. General: Have the authors explored annealing samples to reduce inhomogeneities and improve transmittance?

Not yet, maybe in a future work.….

Minor points:

  • Line 14: Should be “means”, not “mean”.
  • Lines 14-16: Sentence starting “Moreover…” Check grammar.
  • Line 52-53: Should be “preforms” not “performs”.
  • Line 73: I think it should just be “viscosity as a function of temperature”, not “viscosity behavior”.
  • Line 142: Should be “meters of fiber”.

Corrected the text

Thank you for your review and your comments

Reviewer 3 Report

Re: Investigation on chalcogenide glass additive manufacturing for shaping mid-infrared optical components and microstructured optical fibers

By Julie Carcreff et al.

Although additive manufacturing looks very attractive in application to glasses, it is still in the phase of infancy. The authors confess themselves that their printed objects are far from being ideal. Nevertheless, I believe that the obtained results on As30Se50Te20 (TAS) composition are an important step towards 3D printing of glassy structures. The authors honestly analyzed the problems (like impurities, layer boundaries, trapped bubbles, etc.; I would also add here thermal tensions) and correctly identified the routes of how to improve the process. I think the paper is worth of publication after minor revisions:

  • On page 5 the authors claim that “the Tg of the printed glass and the initial glass are similar.” I would prefer to see a DSC thermogram for the printed glass (for example, embedded in Fig. 1a to compare with DSC of initial glass).
  • The extrusion temperature is close to 300 C. So, it would be nice if DSC curves presented in Fig. 1a were also cut at 300 C or slightly above.
  • The quality of Fig. 7a and 8b can be improved (at least to see scale bar numbers on the ruler).
  • Proofreading is recommended.

Author Response

Although additive manufacturing looks very attractive in application to glasses, it is still in the phase of infancy. The authors confess themselves that their printed objects are far from being ideal. Nevertheless, I believe that the obtained results on As30Se50Te20 (TAS) composition are an important step towards 3D printing of glassy structures. The authors honestly analyzed the problems (like impurities, layer boundaries, trapped bubbles, etc.; I would also add here thermal tensions) and correctly identified the routes of how to improve the process. I think the paper is worth of publication after minor revisions:

  • On page 5 the authors claim that “the Tg of the printed glass and the initial glass are similar.” I would prefer to see a DSC thermogram for the printed glass (for example, embedded in Fig. 1a to compare with DSC of initial glass).

The DSC curve of a printed glass has been added in fig 1.

  • The extrusion temperature is close to 300 C. So, it would be nice if DSC curves presented in Fig. 1a were also cut at 300 C or slightly above.

New DSC curves has been added until 300 °C for the initial and the printed TAS glasses

  • The quality of Fig. 7a and 8b can be improved (at least to see scale bar numbers on the ruler).

We do not succeed to improve the figure 7. However, we have changed and improved the quality of the figure 8

  • Proofreading is recommended.

Some typos have been found and corrected in the text.

Thank you for reviewing our article. Thank you for your constructive comments